# Liubao Tea Extract Attenuates High-Fat Diet and Streptozotocin-Induced Type 2 Diabetes in Mice by Remodeling Hepatic Metabolism and Gut Microbiota

**DOI:** 10.3390/nu17162665

**Published:** 2025-08-18

**Authors:** Jichu Luo, Zhijuan Wei, Yuru Tan, Ying Tong, Bao Yang, Mingsen Wen, Xuan Guan, Pingchuan Zhu, Song Xu, Xueting Lin, Qisong Zhang

**Affiliations:** 1Guangxi Key Laboratory of Special Biomedicine, School of Medicine, Guangxi University, Nanning 530004, China; luojichu1999@163.com (J.L.); weizhijuan2022@163.com (Z.W.); 15894549320@163.com (Y.T.); tongying0206@163.com (Y.T.); q954637097@163.com (M.W.); guanxuan010203@163.com (X.G.); fighting68_xusong@163.com (S.X.); kyl05170606@163.com (X.L.); 2Hubei Provincial Key Laboratory of Occurrence and Intervention of Rheumatic Diseases, Hubei Minzu University, Enshi 445000, China; ybsept@hbmzu.edu.cn; 3State Key Laboratory for Conservation and Utilization of Subtropical Agro-Bioresources, Guangxi University, Nanning 530004, China; zhupingchuan12@163.com; 4Center for Instrumental Analysis, Guangxi University, Nanning 530004, China

**Keywords:** type 2 diabetes, Liubao tea extract, hepatic metabolism, gut microbiota, short-chain fatty acids

## Abstract

**Background**: Type 2 diabetes (T2D) has become a serious global public health concern. Liubao tea (LBT) has demonstrated beneficial effects on gut microbiota and glucose-lipid metabolism, holding promising therapeutic potential for T2D; however, its underlying mechanisms remain unclear. This study aims to elucidate the potential mechanisms of Liubao tea extract (LBTE) against T2D. **Methods**: LC-MS technology was used to identify the chemical components of LBTE and combined with network pharmacology and molecular docking to screen its potential active ingredients and targets for improving T2D. Therapeutic efficacy was assessed in high-fat diet/streptozotocin (HFD/STZ)-induced diabetic mice via serum biochemical analyses and histopathological examinations. Serum metabolomics, 16S rRNA sequencing, quantification of short-chain fatty acids (SCFAs), quantitative real-time PCR (qPCR), and antibiotic-treated pseudo-germ-free models were employed to elucidate the underlying mechanisms. **Results**: LBTE effectively reduced blood glucose levels and improved lipid metabolism, primarily by promoting hepatic glycogen synthesis and suppressing glycerophospholipid synthesis. LBTE also alleviated hepatic inflammation by modulating inflammatory cytokine expression. Additionally, LBTE reshaped the gut microbiota profiles by decreasing harmful bacteria and increasing SCFA-producing bacteria, resulting in elevated fecal SCFAs. SCFAs contributed to improving hepatic metabolism and inflammation, enhancing intestinal barrier function. Notably, these effects were abolished by antibiotic-induced microbiota depletion, confirming the microbiota-dependent mechanism of LBTE. Quercetin, luteolin, genistein, and kaempferol were considered as potential active ingredients contributing to the antidiabetic effects of LBTE. **Conclusions**: These findings provide novel perspectives on the viability of LBTE as a complementary strategy for T2D prevention and management.

## 1. Introduction

Diabetes has become a critical global health concern [1], with current projections estimating that the global prevalence will reach 783.2 million by 2045, of which approximately 90% are expected to be type 2 diabetes (T2D) cases [2]. T2D is a multifactorial metabolic disorder characterized by insulin resistance and compromised insulin secretion, resulting in chronic hyperglycemia and disrupted glycometabolism. Current therapeutic strategies primarily rely on pharmacological approaches, which often necessitate long-term use and may result in gastrointestinal complications and hepatorenal toxicity [3]. Therefore, the development of safer and more effective alternatives for T2D management remains an urgent priority.

Tea (*Camellia sinensis* (L.) Kuntze) has a long history of medicinal use in China, traditionally attributed to the legend of Shen-Nong tasting herbs. Its therapeutic applications have evolved over thousands of years and are well documented in classical medical texts and traditional Chinese pharmacopeias [4,5]. Among the various tea types, dark tea, a post-fermented tea characterized by its unique organoleptic attributes, has been traditionally used in medicinal formulations for the prophylaxis and management of obesity and diabetes [6]. Liubao tea (LBT), a representative Chinese dark tea with a history of over 1500 years, is a nationally recognized geographical indication product originating from Wuzhou, Guangxi [7,8]. Contemporary pharmacological research has revealed that LBT is rich in polysaccharides, polyphenols, and flavonoids, exhibiting hypolipidemic, hepatoprotective, and hypoglycemic bioactivities [9]. Our previous studies also found that LBT ameliorated non-alcoholic fatty liver disease (NAFLD) and hyperlipidemia by modulating lipid metabolism and gut microbiota [10,11]. Although preliminary evidence indicates that LBT may alleviate T2D symptoms [12,13], its bioactive constituents and underlying mechanisms remain largely unclear.

The gut microbiota represents a diverse and dynamic microbial community that is essential for regulating host metabolic balance. Among its metabolites, short-chain fatty acids (SCFAs) are key mediators that modulate glucose and lipid metabolism through activation of G-protein-coupled receptors (GPCRs) and downstream hepatic signaling pathways [14]. Since the introduction of the microbiome into the biomedical field, the host metabolism mediated by gut microbiota has been recognized as a key factor in the pathogenesis and treatment of metabolic diseases, including diabetes [15]. (−)-Epicatechin has been found to modulate the gut microbiota and liver insulin signaling pathways to alleviate T2D [16]. Similarly, *L. gasseri CKCC1913* may exert antidiabetic effects by increasing the abundance of beneficial bacteria that produce SCFAs, some of which can reach the liver and contribute to glucose and lipid homeostasis [17]. These findings underscore the critical role of the gut microbiota and its mediated metabolism in T2D treatment. However, whether LBT can effectively improve T2D by modulating the gut microbiota and hepatic metabolism has not yet been reported, and its underlying mechanisms remain largely unexplored.

Given the complexity of metabolic disease pathogenesis, the integration of bioinformatics and multiomics has emerged as a powerful strategy to elucidate drug mechanisms and disease pathogenesis [10]. Bioinformatics serves as an important computational tool for analyzing gene expression data and identifying molecular targets and mechanisms involved in various diseases [18]. Multiomics analysis, such as metabolomics and gut microbiota profiling, offers comprehensive approaches for identifying natural product targets and elucidating their pharmacological mechanisms from multiple perspectives [19]. In particular, network pharmacology and molecular docking have been widely used to predict the bioactive components of natural products, potential targets of action, and the mechanisms of their interactions [20]. Combining metabolomics with these approaches facilitates the identification of critical metabolites and pathways involved in disease regulation [21]. Additionally, modulations in gut microbiota composition have been demonstrated to exert significant effects on host metabolic profiles and metabolic pathways [22]. Thus, the integration of bioinformatics with multiomics holds significant promise for deepening our understanding of the mechanisms by which LBT exerts its therapeutic effects in T2D.

This study aims to elucidate the underlying mechanisms by which Liubao tea extract (LBTE) exerts its antidiabetic effects, with a particular focus on the modulation of gut microbiota and microbiota-mediated metabolic pathways. The chemical composition of LBTE was preliminarily characterized through LC-MS analysis. Based on the identified constituents, network pharmacology and molecular docking analyses were employed to predict the active components, therapeutic targets, and key signaling pathways involved in its antidiabetic activity. Selected active compounds were subsequently validated using chemical standards. To assess the therapeutic efficacy of LBTE, a T2D mouse model was established through high-fat diet/streptozotocin (HFD/STZ) induction. Then, the effects of LBTE treatment were evaluated by serum biochemical analyses and histopathological examination of major tissues. Untargeted metabolomics was conducted to identify key metabolic alterations following LBTE intervention. Meanwhile, 16S rRNA gene sequencing was used to profile the gut microbiota composition and identify key genera modulated by LBTE. To further explore the underlying mechanisms, quantitative real-time PCR (qPCR) was conducted to assess the expression of genes involved in metabolic and inflammatory pathways. Additionally, pseudo-germ-free mouse models were utilized to validate the microbiota-dependent effects of LBTE. Collectively, this study is expected to provide novel insights into anti-diabetes mechanisms of LBTE and support its potential as a promising complementary strategy for the prevention and management of T2D.

## 2. Materials and Methods

### 2.1. Chemicals and Reagents

MS-grade methanol, water, acetonitrile, and formic acid were purchased from Merck & Co. (Billerica, MA, USA). LBT was provided by China Tea Co., Ltd. (Wuzhou, Guangxi, China) (Batch number: S002/2022), conforming to GB/T 32719.4-2016 (Brick LPT, Second class) [23]. Standards of quercetin (C15H10O7, HPLC > 99%), luteolin (C15H10O6, HPLC > 99%), genistein (C15H10O5, HPLC > 99%), caffeine (C8H10N4O2, HPLC > 99%), naringenin (C15H12O5, HPLC > 99%) and kaempferol (C15H10O6, HPLC > 99%) were sourced from Chengdu RefMedic Biotech Co., Ltd. (Chengdu, China). Streptozotocin (STZ), metformin hydrochloride, and citrate buffer were obtained from Macklin Co., Ltd. (Shanghai, China). All other chemicals were analytical grade or higher.

### 2.2. Preparation and Preliminary Chemometric Analysis of LBTE

LBTE was prepared using a modified version of a previously reported method [24], with detailed procedures provided in the Appendix A. In accordance with national analytical standards, major components including total polyphenols (GB/T 8313-2018) [25], total flavonoids (SN/T 4592-2016) [26], total free amino acids (GB/T 8314-2013) [27], and total polysaccharides (NY/T 1676-2008) [28], were quantified using validated chemometric protocols. The total soluble sugar content was determined using the anthrone-sulfuric acid colorimetric method, with glucose as the calibration standard.

### 2.3. Characterization of the Chemical Composition of LBTE

LBTE (1.5 mg) was dissolved in 1.5 mL of methanol and subsequently filtered through a microporous membrane (0.22 μm) to prepare the test solution. The prepared solution was analyzed and characterized using a Waters ACQUITY UPLC I-Class PLUS System (Waters, Milford, MA, USA) (SN: L22BSP766G) coupled with a SELECT SERIES Cyclic IMS (Milford, MA, USA) (SN: GBC086). Detailed LC-MS analytical parameters are presented in Appendix A.

### 2.4. Network Pharmacology Analysis

#### 2.4.1. Drug Target Prediction and Disease Target Identification

Active ingredients in LBTE were screened using the SwissADME database. Potential targets of the active ingredients were predicted using the SwissTargetPrediction database. T2D-related potential targets were obtained by collecting and de-duplicating data from the OMIM and GeneCards databases. The overlapping targets between the active ingredients and T2D were identified and visualized using a Venn diagram. Protein–protein interaction (PPI) analysis was performed using the STRING database and visualized in Cytoscape 3.9.1. Gene Ontology (GO) and Kyoto Encyclopedia of Genes and Genomes (KEGG) enrichment analyses were performed using the Metascape platform. The top 10 GO terms and top 20 KEGG pathways were visualized using the Wei Sheng Xin cloud platform.

#### 2.4.2. Molecular Docking

The 2D structures of the core compounds were sourced from the PubChem database and energy-minimized using the Molecular Operating Environment (MOE, version 2019.0102). The crystal structures of key proteins were retrieved from the RCSB PDB database, including AKT1 (PDB ID: 1UNQ; resolution: 0.98 Å), TNF (PDB ID: 5UUI; resolution: 1.40 Å), and GSK3B (PDB ID: 1O6L; resolution: 1.60 Å). Detailed information regarding the molecular docking procedures is provided in the Appendix A.

### 2.5. Identification of Potential Active Ingredients in LBTE

Standards of four ingredients (quercetin, luteolin, genistein, and kaempferol) were analyzed using the same LC-MS conditions as those applied for the chemical profiling of LBTE. Each component was verified based on retention time and fragmentation information. Detailed LC-MS analytical parameters are provided in Appendix A.

### 2.6. Quantitative Analysis of Potential Ingredients in LBTE

LBTE (1.5 mg) was dissolved in 1.5 mL of methanol and subsequently filtered through a microporous membrane (0.22 μm) to prepare the test solution. The sample was analyzed using an Agilent 1260 Infinity LC system (SN: DEAEQ48900) and an Agilent 6545 Q-TOF mass spectrometer (SN: SG22202E101), equipped with an ESI source (Santa Clara, CA, USA). The analytical conditions of the LC and MS equipment are provided in Appendix A. Standards of six ingredients were analyzed using instrumental methods consistent with those used for analysis of LBTE. Each compound was quantified based on retention time and exact *m*/*z* values using external calibration curves derived from chemical standards.

### 2.7. Experimental Verification

#### 2.7.1. Animal Model and Treatment

Six-week-old male Kunming (KM) mice (18–22 g) were purchased from SPF (Beijing) Biotechnology Co., Ltd. (Beijing, China) (license No. 110324230101502642). Mice were randomly divided into groups according to the RAND function in Microsoft Excel. The individual mouse was considered the experimental unit within the studies. All experimental mice were housed with free access to food and water under the controlled conditions of 25 ± 2 °C temperature, 50 ± 5% relative humidity, and a 12 h light/dark cycle.

A previously reported protocol for T2D induction was followed with minor modifications [29,30]. The design of Experiment 1 is illustrated in Appendix A. After one week of acclimatization, thirty male mice were randomly assigned to five groups (*n* = 6 per group): normal control (NC) group, model control (MC) group, positive control (PC) group, LBTE low-dose (LBTE-L) group, and LBTE high-dose (LBTE-H) group. Except for the NC group (normal-chow diet), all mice from other groups were fed a high-fat diet (HFD) for four weeks, followed by two intraperitoneal injections of STZ (50 mg/kg), and the NC group received equivalent volumes of citrate buffer. Three days after the final injection, mice with fasting blood glucose (FBG) ≥ 11.1 mmol/L were classified as T2D mice [31]. Subsequently, mice in the LBTE-L and LBTE-H groups were orally administered LBTE at doses of 200 and 400 mg/kg, respectively, once daily for three weeks, based on previous studies with minor modifications [11,32,33]. The PC group received metformin (200 mg/kg), while the NC and MC groups received an equivalent volume of normal saline.

To investigate the microbiota-dependent effects of LBTE, a second animal experiment (Experiment 2) was conducted based on a previously described antibiotic depletion protocol with minor modifications [34]. The experimental design is depicted in Appendix A. Briefly, eighteen T2D mice were randomly assigned to three groups (*n* = 6 per group): MC group, MC + Antibiotic mixture (MA) group, and LBTE + Antibiotic mixture (LBTEA) group. The mice in the MA and LBTEA groups were orally administered a mixture of antibiotics (ampicillin 130 mg/kg; metronidazole 150 mg/kg; neomycin 100 mg/kg; and vancomycin 130 mg/kg) daily, and mice in the MC group were orally equivalent volume of sterile water. Additionally, mice in the LBTEA group were treated with LBTE (400 mg/kg), whereas the other groups received an equal volume of normal saline. FBG levels, body weight, food intake, and water intake were monitored weekly during the treatment period.

#### 2.7.2. Sample Collection

At the end of the experiment, all mice were euthanized, and blood samples were collected immediately. Serum was isolated by centrifugation at 5000 rpm for 8 min at 4 °C and subsequently stored at −80 °C for further analysis. Major organs and tissues were carefully excised, rinsed in cold saline, blotted dry, and weighed before being stored or processed for histological analysis.

#### 2.7.3. Biochemical Index Determination

Serum levels of alanine aminotransferase (ALT), aspartate transaminase (AST), total bile acid (TBA), creatinine (Cr), uric acid (UA), total cholesterol (TC), triglycerides (TG), low-density lipoprotein cholesterol (LDL-C) and high-density lipoprotein cholesterol (HDL-C) were measured using commercial kits from Nanjing Jiancheng Bioengineering Institute (Nanjing, China), according to the manufacturer’s instructions.

#### 2.7.4. Histopathological Analysis

Tissue samples from the liver, adipose tissue, heart, lung, kidney, and colon were fixed in 4% paraformaldehyde, dehydrated, washed, embedded in paraffin, sectioned (4 μm), and baked. Hematoxylin and eosin (H&E) staining was performed on all tissue sections, while liver sections were additionally stained with Oil Red O to evaluate lipid accumulation. Histopathological changes were observed under a light microscope.

### 2.8. Metabolomic Analysis of Serum Samples

Serum sample preparation followed our previous study [11], using tamoxifen and cholic acid-d4 as internal standards. To ensure the stability and reproducibility of data acquisition, quality control (QC) samples were prepared by mixing equal aliquots (5 μL) of each sample. Untargeted metabolomic analysis of serum samples was conducted using an Agilent 1260 Infinity LC system (SN: DEAEQ48900) and an Agilent 6545 Q-TOF mass spectrometer (SN: SG22202E101), equipped with an ESI source (Santa Clara, CA, USA). The LC and MS conditions used in the metabolomic study are detailed in Appendix A.

### 2.9. Gut Microbiota Analysis

Bacterial DNA extraction from fecal samples in mice, PCR amplification, and sequencing were performed according to the standard procedure of Novogene Co., Ltd. (Beijing, China), as previously described [35]. Detailed experimental protocols and analytical pipelines are provided in the Appendix A.

### 2.10. MetOrigin Analysis

The MetOrigin platform was employed for traceability analysis of differential metabolites and integrative analysis of the gut microbiome and serum metabolome [36]. Specifically, MetOrigin’s deep analysis module was applied for origin-based metabolite pathway enrichment analysis (MPEA) and Bio-Sankey network visualization, both accessible on the official website.

### 2.11. SCFA Analysis

SCFAs in fecal samples were extracted and derivatized according to a previously reported method [37]. The preparation process of SCFAs and conditions for GC-MS analysis are detailed in the Appendix A. Quantification of SCFAs was performed using external calibration curves generated from peak areas of SCFA standards.

### 2.12. Determination of Gene Expression Using qPCR

Total RNA was extracted from liver and colon tissues using the TransZol Up (TransGen Biotech, Shanghai, China) reagent and reverse-transcribed into cDNA using the SweScript All-in-One RT SuperMix (Servicebio, Wuhan, China). Then, PCR reactions were conducted using the MonAmp SYBR Green qPCR Mix (Monad Biotech, Shanghai, China), with GAPDH serving as an internal reference. The gene primers are listed in Appendix A. The mRNA expression was calculated using the 2^−ΔΔCT^.

### 2.13. Statistical Analysis

The metabolomic data were processed according to the methodology in our previous study [11]. Data are expressed as the mean ± standard deviation (SD). Statistical comparisons between groups were performed using Student’s *t*-test, with *p* < 0.05 considered statistically significant. Graphs were generated using GraphPad Prism 9.

## 3. Results

### 3.1. Chemical Composition Profiling of LBTE

To characterize the primary chemical constituents of LBTE, the levels of total polysaccharides, total polyphenols, total flavonoids, total free amino acids, and total soluble sugars were quantitatively assessed. Among these components, total flavonoids and total polyphenols exhibited notably high content (Appendix A), indicating that these constituents may significantly contribute to the therapeutic potential of LBTE. The total ion chromatogram (TIC) of LBTE is presented in Appendix A, providing the comprehensive chemical fingerprint of LBTE. A total of 34 compounds were preliminarily identified by comparison with the database, including alkaloids, catechins and flavonoids, with flavonoids representing the largest proportion of 30 compounds. The detailed information on these compounds is provided in Appendix A.

### 3.2. Network Pharmacology Analysis and Potential Active Ingredients Identification

To elucidate how these components exert antidiabetic effects, network pharmacology analysis was performed. A total of 140 potential targets associated with LBTE were identified from the SwissTargetPrediction database. Concurrently, 2051 targets related to T2D were obtained from the disease-related databases. After crossing these targets, 62 overlapping targets were found, suggesting their potential involvement in the antidiabetic effects of LBTE (Figure 1A). Core compounds were selected through network topology analysis based on degree centrality values and included quercetin, kaempferol, luteolin, genistein, etc. Subsequent analysis of the 62 intersection targets identified the top 10 hub genes based on degree values: AKT1, TNF, EGFR, SRC, ESR1, PTGS2, MMP9, GSK3B, KDR, and IGF1R (Figure 1B). These hub genes were considered key therapeutic targets for LBTE in the treatment of T2D. To elucidate the biological relevance of these targets, GO and KEGG pathway enrichment analyses were conducted, and the top 10 GO terms and top 20 KEGG pathways are shown in Figure 1D,E. Notably, the PI3K-Akt signaling pathway, insulin resistance, and MAPK signaling pathway emerged as key pathways implicated in T2D pathogenesis, with AKT1, TNF, and GSK3B identified as central regulatory nodes. Additionally, a core compound-target-pathway network was constructed to visualize the multi-target and multi-pathway interactions of the core compounds (Figure 1C). The network revealed that each core compound potentially regulates multiple targets, collectively contributing to the modulation of metabolic and signaling pathways relevant to T2D.

Subsequently, four potential active ingredients including quercetin, luteolin, genistein, and kaempferol in LBTE were successfully identified by matching their exact mass, mass error, retention time, and fragmentation patterns with those of chemical standards analyzed under identical conditions (Appendix A).

Based on the network pharmacology results, AKT1, TNF, and GSK3B were identified as key targets and subsequently selected for molecular docking analysis. As summarized in Appendix A, the docking scores (S score) and predicted binding interactions between the core compounds (quercetin, luteolin, genistein, and kaempferol) and the target proteins are presented. The docking results demonstrated favorable binding conformations and strong affinities (Appendix A). These findings support the hypothesis that LBTE exerts its antidiabetic effects through interactions with multiple targets, particularly involving AKT1, TNF, and GSK3B.

### 3.3. Quantification of Potential Ingredients in LBTE

Each component was validated by comparing retention time and mass spectral fragmentation data of chemical standards and their contents were quantified using an external calibration curve based on peak area (Figure 2). The concentrations of caffeine, genistein, naringenin, quercetin, luteolin and kaempferol in LBTE were determined to be 30.1 ± 1.8, 1.04 ± 0.05, 0.47 ± 0.07, 1.2 ± 0.02, 0.62 ± 0.12, and 0.76 ± 0.01 mg/g, respectively. These results indicate that LBTE is rich in flavonoids and alkaloids, which may contribute to its antidiabetic effects.

### 3.4. Amelioration of LBTE on T2D Mice

As shown in Figure 3A, the body weight of T2D mice was significantly lower than that of the NC group between weeks 6 and 9. Treatment with LBTE and metformin further reduced body weight compared to the MC group, with a significant reduction observed in the high-dose LBTE group (*p* < 0.05). These results suggest that LBTE may partially exert its metabolic benefits in T2D mice through weight modulation.

While the differences in food and water intake were not statistically significant, a decreasing trend in food consumption in the LBTE-treated groups suggests a possible influence of LBTE on appetite. (Figure 3C,D) FBG levels were notably reduced following LBTE administration, particularly in the high-dose group (*p* < 0.05). Interestingly, FBG levels in the PC group were comparable to those in the LBTE-L group and did not show a significant reduction relative to the MC group (Figure 3B). The comparison of two different administered dose methods revealed nearly identical FBG levels in both groups (Appendix A). Further evaluation of different dosing regimens revealed that triple dosing produced a more pronounced glucose-lowering effect than single dosing (Appendix A), suggesting that hypoglycemic efficacy depends on dosing frequency and drug exposure duration, consistent with the reported elimination half-life (~5 h) of metformin [38], it could be hypothesized that the hypoglycemic efficacy of metformin and LBTE was strongly influenced by dosing frequency and duration of exposure. Serum lipid analyses showed that the levels of TC, TG, LDL-C, and HDL-C were significantly altered in the MC group compared to the NC group (*p* < 0.05) (Figure 3T–W). LBTE treatment significantly reversed these changes (*p* < 0.05).

Histopathological examination revealed that the MC group exhibited abnormal hepatic architecture, lipid droplet accumulation, and inflammatory infiltration compared to the NC group, whereas LBTE markedly ameliorated these pathological changes (Figure 4A,B). Adipose tissue also revealed fat accumulation in the MC group, which was alleviated by LBTE, particularly at high doses (Figure 4C). Consistently, liver weight and index were significantly elevated in the MC group but significantly reduced following LBTE treatment (*p* < 0.05) (Figure 3K,P). The levels of ALT, AST, and TBA were significantly elevated in the MC group compared with the NC group and normalized upon LBTE intervention (*p* < 0.05) (Figure 3E–G), suggesting hepatoprotective effects of LBTE for T2D mice. Moreover, LBTE treatment significantly reversed the changes in kidney weight and index observed in the MC group (*p* < 0.05) (Figure 3N,S). Serum levels of Cr and UA were also markedly elevated in the MC group compared to the NC group, indicating renal dysfunction, while LBTE intervention significantly lowered Cr and UA levels (*p* < 0.05) (Figure 3H,I). Although no obvious pathological changes were observed in kidney tissues across all groups (Figure 4F), the biochemical improvements suggested that LBTE conferred renal protective effects. Additionally, spleen weight and index were significantly elevated in MC group and dose-dependently normalized following LBTE treatment (*p* < 0.05, Figure 3L,Q), suggesting a potential immunomodulatory effect. While no significant changes were observed in lung weight and index (Figure 3M,R), histological analysis revealed that LBTE alleviated alveolar wall thickening and inflammatory infiltration present in T2D mice (Figure 4E), suggesting a protective effect against pulmonary inflammation. Finally, no significant differences in heart weight or index were observed (Figure 3J,O), and no pathological changes were noted in heart tissues across all groups (Figure 4D), suggesting that neither T2D nor LBTE treatment had a significant impact on cardiac morphology during the study period. In the colon, the MC group showed disrupted colonic tissue structure and increased inflammatory infiltration, whereas LBTE treatment effectively restored colon integrity and alleviated the inflammatory state (Figure 4G), highlighting a protective effect on intestinal barrier function. In summary, LBTE exerted multifaceted protective effects in T2D mice, including improvements in glycemic control, lipid profiles, liver and kidney function, as well as attenuation of inflammation and tissue damage across multiple organs, thereby supporting its therapeutic potential in metabolic disease management.

### 3.5. Influence of LBTE Treatment on the Serum Metabolic Profiles of T2D Mice

To evaluate the metabolic impact of LBTE on T2D mice, serum untargeted metabolomics was performed. As shown in Appendix A, the TIC of QC samples exhibited substantial overlap, indicating excellent instrument performance and validating the reliability of the acquired data. Subsequently, an orthogonal partial least squares-discriminant analysis (OPLS-DA) revealed clear clustering and separation across all groups. As expected, samples from the MC group were distinctly separated from those of the NC group, reflecting pronounced metabolic disturbances induced by T2D. Notably, the metabolic profile of the LBTE-H group diverged significantly from the MC group and trended toward the NC group (Figure 5A,B), suggesting a restorative effect of LBTE on T2D-associated metabolic dysfunction. Model validity and predictive accuracy were evaluated using R^2^ and Q^2^ values, and permutation tests (*n* = 200) further confirmed the robustness of the OPLS-DA model and excluded overfitting (Figure 5C,D). Key differential metabolites were screened based on the OPLS-DA models using the following criteria: fold change (FC) > 1.5 or <0.67, variable importance in projection (VIP) > 1, and *p* < 0.05. The differential metabolites were classified into several chemical categories, mainly including amines, prenol lipids, sterol lipids, fatty acids (FAs), glycerophospholipids (GPs), sphingolipids (SPs), glycerolipids, lipids, amino acids (AAs) and xanthines (Figure 5E–G). Among these, GPs accounted for the largest fraction, exceeding 50% of all differential metabolites, followed by FAs and SPs.

To disclose the potential mechanisms underlying T2D pathogenesis and the therapeutic effects of LBTE, metabolic pathway enrichment analysis was conducted based on serum differential metabolites. In T2D mice, obvious dysregulation was observed in both sphingolipid (SP) and glycerophospholipid (GP) metabolism, with statistical thresholds of *p* < 0.05 or impact factor (IF) > 0.1 (Figure 5H). Remarkably, these metabolic disturbances were attenuated following LBTE intervention (Figure 5I,J), suggesting that modulation of these pathways could be involved in the therapeutic effects of LBTE. Further analysis identified 16 shared serum differential metabolites across four groups (Appendix A). To assess their diagnostic and therapeutic relevance, ROC analysis was employed, with stringent screening criteria of AUC > 0.95 and FC > 5 to minimize false positives. Among these, 13 metabolites demonstrated excellent discriminatory power between the NC and MC groups, with AUC values reaching up to 1.00, thereby underscoring their potential as robust diagnostic biomarkers for T2D (Appendix A). ROC analysis of the differential metabolites from comparing the MC group with the LBTE-L and LBTE-H groups identified four and five metabolites, respectively, with AUC values exceeding 0.95 (Appendix A). Notably, metabolites such as PS 38:0, PC 40:7, PE 38:2, PE 41:3, and 1-methylhistamine, mainly derived from GPs and amines, were considered as the key indicators of LBTE’s therapeutic benefits in T2D. Consequently, LBTE exerted its anti-T2D effects primarily through the modulation of GP metabolism.

### 3.6. Regulation of LBTE on Expression of Key Genes Involved in Inflammatory Response, Glucose and Lipid Metabolism Pathways

As depicted in Figure 5K–R, the mRNA expression levels of *IL-10* and *AKT1* were significantly decreased in the MC group compared to the NC group, whereas the expression levels of *IL-6*, *TNF-α*, *GSK3B*, *AGPAT*, *GPAT*, and *PAP* were significantly increased (*p* < 0.05). Notably, the administration of LBTE significantly reversed these alterations, restoring the gene expression profiles close to those of the NC group (*p* < 0.05). The results indicated that LBTE alleviated T2D by suppressing inflammation and modulating key pathways involved in glucose utilization and lipid synthesis, thereby improving metabolic imbalances.

### 3.7. LBTE Administration Modulated the Gut Microbiota Profiles in T2D Mice

To evaluate the impact of LBTE on gut microbial ecology, 16S rRNA sequencing was conducted on fecal samples. A total of 176 amplicon sequence variants (ASVs) were identified across all samples, with 165 ASVs unique to the MC group and 163 ASVs unique to the LBTE-H group (Figure 6B), highlighting distinct microbial signatures between disease and treatment conditions. Analysis of α-diversity indexes, including chao1, observed_species, pielou_e, and shannon indexes, revealed no significant differences between the NC and MC groups as well as between the MC and LBTE groups (Figure 6D–G). However, the α-diversity indexes in the LBTE-H group were more similar to those in the NC group, indicating a partial restoration of microbial richness and evenness following LBTE treatment. β-diversity analysis of gut microbiota, assessed by OPLS-DA (Figure 6I) and hierarchical clustering (Figure 6H), revealed clear distinctions between the NC and MC groups, as well as between the MC and LBTE-H groups, with notable proximity in microbial profiles between the NC and LBTE-H groups. These results indicate that LBTE administration remodeled the gut microbial community structure disrupted by T2D. At the phylum level (Figure 6A), the gut microbiota was predominantly composed of *Firmicutes*, *Bacteroidetes*, *Verrucomicrobiota* and *Patescibacteria*. The MC group exhibited an increased abundance in *Firmicutes* and a decreased abundance in *Bacteroidetes* compared to the NC and LBTE-H groups. LBTE treatment partially reversed this dysbiosis in T2D mice, restoring a microbial composition more similar to healthy mice.

At the genus level (Figure 6C), *Akkermansia*, *Lactobacillus*, *Ligilactobacillus*, and *Bacteroides* were dominant across all groups. The relative abundance of *Lachnospiraceae_NK4A136_group*, *Akkermansia*, *Bacteroides*, *Lactobacillus*, and *Helicobacter* was increased, while the *Ligilactobacillus*, *Candidatus_Saccharimonas*, and *Enterorhabdus* were decreased in the MC group. These alterations were reversed following LBTE-H treatment (Appendix A). Further, Linear Discriminant Analysis Effect Size (LEfSe) analysis with an LDA threshold of three was conducted to reveal specific bacterial changes. The MC group was enriched in *Oscillospirales* at the order level, the NC group in *Micrococcales*, *Chloroplast*, and *Cyanobacteriia*, while the LBTE-H group was characterized by *Erysipelotrichaceae* and *Clostridium_methylpentosum_group* (Figure 6J,K). To assess functional shifts in microbial metabolism, PICRUSt2 was employed for predictive metagenomic analysis. As shown in Appendix A, the top 10 most abundant functional proteins were primarily involved in bacterial material transport and carbohydrate metabolism. Notably, genes encoding K07024 (sucrose-6-phosphatase, SPP) and K15634 (probable phosphoglycerate mutase, gpmB) were down-regulated in the MC group but restored following LBTE treatment, suggesting that LBTE may enhance microbial-mediated carbohydrate metabolism. Collectively, LBTE treatment reshaped the gut microbiota in T2D mice by restoring microbial balance and diversity, enriching beneficial taxa, and enhancing predicted carbohydrate metabolism. These findings highlight the microbiota-mediated mechanisms underlying LBTE’s antidiabetic effects.

### 3.8. MetOrigin Tracing Analysis of Differential Metabolites

In the comparison between the NC and MC groups, 136 differential metabolites associated with T2D were identified (Appendix A). MPEA revealed that these metabolites were involved in 13 co-metabolism pathways (Appendix A). Among them, SP and GP metabolism (log0.05 *p* > 2) were identified as the most significantly enriched pathways in the co-metabolism between gut microbiota and the hosts, indicating their pivotal role in the pathogenesis of T2D (Appendix A). In the GP metabolism-related metabolic reaction R01320, PC 40:7 was identified as the key metabolite, with *Helicobacter* being the primary associated genus, showing a strong positive correlation (Appendix A). Similarly, PS 38:0 was the major metabolite in reaction R01800, which involved four bacterial genera related to GP metabolism. Among these, *Bacteroides*, *Akkermansia*, and *Helicobacter* exhibited positive correlations with PS 38:0, whereas *Ligilactobacillus* showed a negative correlation (Appendix A). In addition, three key metabolites including sphingosine, sphinganine, and phytosphingosine were identified in the metabolic reactions R01494, R06518, and R06528, respectively, related to SP metabolism, underscoring their contribution to lipid dysregulation in T2D (Appendix A).

In the comparison between the MC and LBTE-L groups, 70 differential metabolites were identified (Appendix A). MPEA revealed that nine significantly enriched co-metabolic pathways, with GP metabolism being the most notably modulated (Appendix A). Similarly, in the comparison between the MC and LBTE-H groups, 68 differential metabolites were identified (Appendix A). MPEA indicated that modulation of seven co-metabolic pathways was associated with co-metabolism pathway databases, among which GP metabolism was most significantly affected (Appendix A). Consistently, PC 40:7 and *Helicobacter* remained central elements in metabolic reaction R01320, which showed a positive correlation (Appendix A). Moreover, PS 38:0 and its associated genera (*Bacteroides*, *Akkermansia*, *Helicobacter*, and *Ligilactobacillus*) were repeatedly highlighted in reaction R01800, further supporting their involvement in GP metabolism (Appendix A). Therefore, LBTE exerted its anti-T2D effects primarily by modulating GP metabolism, likely mediated by restructuring the abundance of key microbiota, thereby restoring host-microbiota metabolic balance to alleviate T2D.

### 3.9. LBTE Enhanced the Intestinal Barrier Function by Promoting SCFA Production and Regulating GPCR Gene Expression

To evaluate the impact of LBTE on gut microbial metabolism, fecal SCFAs were quantitatively analyzed. The levels of three major SCFAs, as well as the total SCFA content, were significantly elevated in the NC and LBTE-treated groups compared to the MC group (*p* < 0.05) (Figure 7A), suggesting that LBTE facilitated the recovery of gut microbial metabolic activity impaired by T2D. In the MC group, the mRNA expression levels of *GPR41* and *GPR109A* in both colon and liver tissues were significantly down-regulated compared to the NC group, whereas LBTE treatment significantly up-regulated the expression of these receptors, with more pronounced effects observed in the high-dose group (*p* < 0.05) (Figure 7D–G). In addition, the mRNA expression levels of tight junction proteins, *ZO-1* and *Occludin*, were significantly lower in the MC group compared to the NC group, indicating compromised intestinal barrier function. LBTE treatment significantly restored the mRNA expression levels of these proteins, thereby effectively improving intestinal barrier function (*p* < 0.05) (Figure 7B,C). Collectively, these results demonstrate that LBTE exerted protective effects in T2D mice by promoting SCFA production and enhancing the expression of GPCRs and tight junction proteins, thus contributing to the restoration of intestinal barrier function.

### 3.10. Correlation Analysis

Correlation analyses were conducted to elucidate the interactions among T2D-related biochemical indicators, key metabolites, key bacterial genera, and SCFA levels. As shown in Figure 7I, the relative abundances of *Helicobacter* showed a significant positive correlation with those of PE 38:2, PE 41:3, PS 38:0, and PC 40:7 (*p* < 0.05). Moreover, the relative abundances of *Bacteroides* exhibited significant positive correlations with the levels of LDL-C and ALT (*p* < 0.05). The relative abundances of *Helicobacter* exhibited significant positive correlations with the levels of TBA and liver weight, while presenting significant negative correlations with the levels of HDL-C (*p* < 0.05). However, the *Ligilactobacillus*, *Candidatus_Saccharimonas*, and *Enterorhabdus* showed the opposite trends, displaying negative correlations with these metabolites and biochemical indicators. As shown in Figure 7H, the levels of SCFAs were negatively correlated with the relative abundances of *Helicobacter*, *Akkermansia*, and *Bacteroides*, while positively correlated with the relative abundances of *Candidatus_Saccharimonas*, *Enterorhabdus*, and *Ligilactobacillus*. Moreover, SCFA levels were negatively correlated with all T2D-related biochemical indicators, except HDL-C, which showed a positive correlation. These findings further suggest that gut microbiota contribute to the amelioration of metabolic disorders and liver dysfunction in T2D by enhancing SCFA production.

### 3.11. LBTE Alleviated T2D Pathology Depending on Gut Microbiota

To confirm whether gut microbiota was involved in the improvement effects of LBTE on T2D, a pseudo-germ-free mouse model was established using a cocktail of broad-spectrum antibiotics. After antibiotic treatment, the α-diversity indexes, including chao1, observed_species, pielou_e, and shannon of both the MA and LBTEA groups, were significantly lower and close to zero compared to the MC group (*p* < 0.05), confirming the successful establishment of the pseudo-germ-free mouse model (Figure 8A–D). NMDS analysis further disclosed that the gut microbiota compositions of the MA and LBTEA groups were highly similar yet distinctly separated from that of the MC group (Figure 8E).

In addition, antibiotic intervention significantly altered the microbial community, transforming it from a highly diverse community to one dominated by *Proteobacteria*. Compared with the MC group, the abundance of *Firmicutes* and *Bacteroidota* significantly decreased, while *Proteobacteria* sharply increased in the MA and LBTEA groups (Figure 8F). Importantly, LBTE treatment failed to restore the gut microbiota profiles in the presence of antibiotic intervention, suggesting a loss of its therapeutic efficacy in the absence of a functional microbiota. We further evaluated the impact of these microbiota changes on SCFA production. As shown in Appendix A, propionic acid levels were significantly lower in the MA and LBTEA groups compared to the MC group (*p* < 0.05), whereas butyric acid and isovaleric acid were almost undetectable. Histological examination revealed that, similar to the MC group, mice in both the MA and LBTEA groups exhibited crypt damage and inflammatory cell infiltration in colonic tissues (Appendix A). The inability of LBTE to ameliorate colonic histopathological abnormalities was closely associated with the reduced SCFA levels following antibiotic treatment.

Regarding physiological parameters, the body weight of mice in the MC group continued to decrease throughout the experiment, and body weight trends in the MA and LBTEA groups mirrored those of the MC group, with no significant differences observed (Figure 8G). Meanwhile, the FBG levels were persistently elevated across all three groups, with no significant differences between them, indicating that the hypoglycemic effect of LBTE was abolished after antibiotic intervention (Figure 8H). Although there was no significant difference in water intake among the three groups of mice (Figure 8J), the food intake in the MA and LBTEA groups decreased compared to the MC group (Figure 8I), likely due to the profound disruption of gut microbiota. There were no significant differences in TC, TG, LDL-C, and HDL-C levels between the MA group and the LBTEA group compared to the MC group (Figure 8K–N). This indicated that the modulatory effects of LBTE on lipid metabolism disorders were abolished after gut microbiota disruption.

Histopathological analysis of adipose tissues further confirmed these findings, revealing that antibiotic intervention reversed the LBTE improvements in lipid accumulation induced by T2D (Figure 8V). Similarly, no significant differences were observed in the liver weight, liver index, serum ALT, AST, and TBA levels between the MA, LBTEA, and MC groups (Figure 8O–S). H&E staining and Oil Red O staining of liver tissues showed that the liver tissues of mice in the MC group showed significant lipid accumulation and inflammation, whereas antibiotics combined with LBTE failed to alleviate liver injury in mice with T2D (Figure 8T,U). Therefore, antibiotic intervention effectively negated the therapeutic benefits of LBTE, resulting in the loss of its ability to improve metabolic disorders, reduce lipid accumulation, and alleviate liver damage. These findings underscore that the therapeutic efficacy of LBTE against T2D is strongly microbiota-dependent, reinforcing the importance of gut-host interactions in metabolic disease modulation.

## 4. Discussion

This study revealed the potential mechanism of LBTE in improving T2D by integrating bioinformatics analysis, multi-omics technology, and in vivo experimental verification. Network pharmacology combined with targeted metabolomics revealed that quercetin, luteolin, genistein, and kaempferol were potential bioactive compounds in LBTE, exerting antidiabetic effects by regulating signaling pathways such as PI3K-Akt, insulin resistance, and MAPK. Molecular docking demonstrated that these flavonoids have strong binding affinities to key targets such as AKT1, GSK3B, and TNF-α. Among them, luteolin and quercetin have been shown to improve the disorders of glucose and lipid metabolism [39,40], genistein has both hypoglycemic and insulin resistance-reducing properties [17], and kaempferol reduces the inflammatory response by inhibiting the NF-κB pathway [41], which provided the theoretical basis for the subsequent in vivo experiments.

In order to investigate the ameliorative effect of LBTE in vivo, studies were conducted in T2D mice, which exhibit T2D-related characteristics consistent with previous studies [42]. In the T2D mouse model, LBTE significantly reduced FBG and improved dyslipidemia, while reversing hepatic lipid accumulation and inflammation, alveolar wall thickening and colonic tissue intestinal barrier damage. Although no significant differences in food intake were observed among the groups, a downward trend in LBTE-treated groups may indicate subtle appetite regulation. Tea polyphenols, including quercetin and kaempferol, have been reported to modulate gut-brain axis activity through the microbiota and its metabolites, which may in turn affect satiety and feeding behavior [43,44]. These findings imply that LBTE may exert mild anorexigenic effects as part of its overall metabolic modulation. In addition, metabolomics further revealed that LBTE exerted its ameliorative effects mainly through the regulation of SP and GP metabolic pathways. Previous studies have highlighted the crucial role of SPs in causing glucose disturbances, as well as abnormal GP metabolism as a factor in the pathophysiology of diabetes [45,46]. The research findings were consistent with these studies, with LBTE being particularly critical in regulating GP metabolism. T2D is a chronic metabolic disease characterized by hyperglycemia and disorders of lipid metabolism [47]. LBTE promoted the conversion of glucose to glycogen in the liver by up-regulating the expression of *AKT1* and down-regulating the expression of *GSK3B*, thereby reducing blood glucose. Meanwhile, glucose undergoes the glycolysis pathway to produce Glycerol-3-phosphate (G-3-P), which is a substrate in GP synthesis pathway [48]. LBTE could block the metabolic flow of G-3-P to diacylglycerol (DAG) by down-regulating the expression of related genes involved in the GP synthesis pathway (including *GPAT*, *AGPAT* and *PAP*) [49], thereby reducing the synthesis and accumulation of GP and TG in the liver, and improving dyslipidemia. Metabolic disorders can lead to abnormal inflammation [47], which is often accompanied by lipid accumulation, thereby exacerbating liver injury [50,51]. LBTE significantly reduced the expression levels of *TNF-α* and *IL-6*, while increasing the expression level of *IL-10* in the liver. The protective effect of LBTE on the liver may be related to regulating metabolic disorders and alleviating inflammatory states.

Studies have shown that alterations in the composition of gut microbiota affect the onset of diabetes [52]. The MC group exhibited a higher *Firmicutes* to *Bacteroidetes* (F/B) ratio compared to other groups, which contributed to metabolic disorders and insulin resistance [53]. LBTE reversed the F/B ratio, enriching beneficial bacteria such as *Candidatus_Saccharimonas* and *Ligilactobacillus*, while inhibiting the proliferation of harmful bacteria including *Bacteroides*, *Akkermansia*, and *Helicobacter*. *Bacteroides* are linked to increased LPS levels and decreased insulin sensitivity, and *Akkermansia* are enriched in T2D samples [54,55]. An increase in *Helicobacter* promotes insulin resistance, accelerating diabetes progression. *Ligilactobacillus* helps improve the function of the intestinal microbiota and intestinal barrier as well as promote carbohydrate metabolism [56]. *Candidatus_Saccharimonas*, an acid-producing bacterium, is important for maintaining intestinal pH [57]. Previous studies have suggested that the gut microbiota could reshape host metabolism [58]. LBTE regulates GP metabolism by modulating the levels of certain beneficial and harmful gut bacteria. The specific resident bacteria in the gut can metabolize indigestible carbohydrates into various SCFAs molecules [59]. Enzymes like SPP and gpmB play crucial roles in bacterial carbohydrate metabolism [60]. The increased carbohydrate metabolism observed in the LBTE-H group reflected an increase in carbohydrate-utilizing bacteria such as *Candidatus_Saccharimonas* and *Ligilactobacillus*.

As metabolites of intestinal flora, SCFAs are produced in the colon and partially enter the liver through the portal vein [61]. SCFAs help maintain intestinal barrier integrity and regulate host inflammatory responses. LBTE significantly increased the content of fecal SCFAs; among them, butyric acid has been widely proven to enhance intestinal barrier function by up-regulating the expression levels of tight junction proteins *ZO-1* and *Occludin* [55]. T2D is usually characterized by insulin resistance, mainly caused directly or indirectly by increased inflammation [62]. SCFA reduces the release of pro-inflammatory factors such as TNF-α, IL-6, and IL-1β by activating signaling pathways mediated by GPR41 and GPR109A receptors [63], thereby synergistically alleviating inflammatory responses and reducing liver injury [64]. Inhibiting inflammation is an important strategy for LBTE treatment of liver injury. Meantime, inflammation can disrupt the insulin signaling pathway and mediate glucose intolerance, with TNF-α further exacerbating metabolic disorders by interfering with the PI3K-Akt signaling pathway [65]. As the core effector molecule of this pathway, AKT promotes glycogen synthesis by inhibiting GSK3B, thereby maintaining blood glucose homeostasis [66]. Elevated levels of serum TG and TC are also important factors in inducing insulin resistance. SCFAs effectively reduce TG and TC levels by inhibiting hepatic adipogenesis and inflammatory cell infiltration [67]. Correlation analysis further confirmed that SCFAs were significantly negatively correlated with lipid indices and liver injury markers, highlighting their metabolic protective effects. The critical role of gut microbiota in mediating the antidiabetic effects of LBTE was further validated in a pseudo-germ-free mouse model. Antibiotic-induced depletion of the gut microbiota abolished the modulatory effects of LBTE on microbial composition, SCFA production, and glucose-lipid metabolism, confirming a microbiota-dependent mechanism.

Nonetheless, this study has certain limitations. The predicted interactions between core compounds (quercetin, luteolin, genistein, and kaempferol) and key protein targets (AKT1, TNF, and GSK3B) were solely based on in silico analyses, without experimental validation. Future studies should employ in vitro and in vivo assays, such as Western blotting or enzyme activity measurements, to substantiate these mechanistic predictions.

## 5. Conclusions

This study demonstrates that LBTE ameliorates T2D through dual regulation of hepatic metabolism and gut microbiota. LBTE promoted glycogen synthesis and suppressed lipid accumulation via regulation of key metabolic genes. Furthermore, LBTE reshaped the gut microbiota to enhance SCFA production, which in turn improved intestinal barrier integrity and alleviated hepatic inflammation and metabolic disturbances via the GPR43/GPR109A-mediated signaling pathway. Notably, these beneficial effects were abrogated upon antibiotic-induced microbiota depletion, confirming a microbiota-dependent mechanism of action. Additionally, quercetin, luteolin, genistein, and kaempferol were identified as the potential active ingredients contributing to the antidiabetic effects of LBTE. Collectively, this study offers new insight into the therapeutic potential of LBTE as a complementary strategy for T2D prevention and management.

## Figures and Tables

**Figure 1 nutrients-17-02665-f001:**
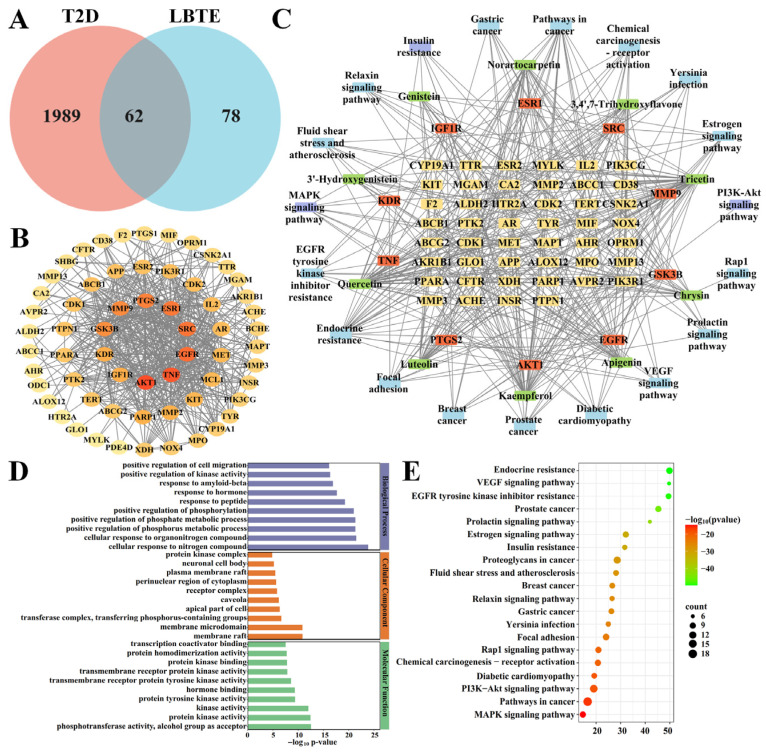
Network pharmacology analysis of Liubao tea extract (LBTE) intervening type 2 diabetes (T2D). (**A**) Venn diagram of the intersection targets of LBTE and T2D. (**B**) PPI (Protein–protein interaction) network. Color depth indicates the target degree. (**C**) Core compounds-targets-pathways network. The light blue, green, purple, yellow, and red indicate KEGG pathways, top 10 compounds, key KEGG pathways, targets, and top 10 targets, respectively. (**D**) GO enrichment analysis. (**E**) KEGG pathway analysis.

**Figure 2 nutrients-17-02665-f002:**
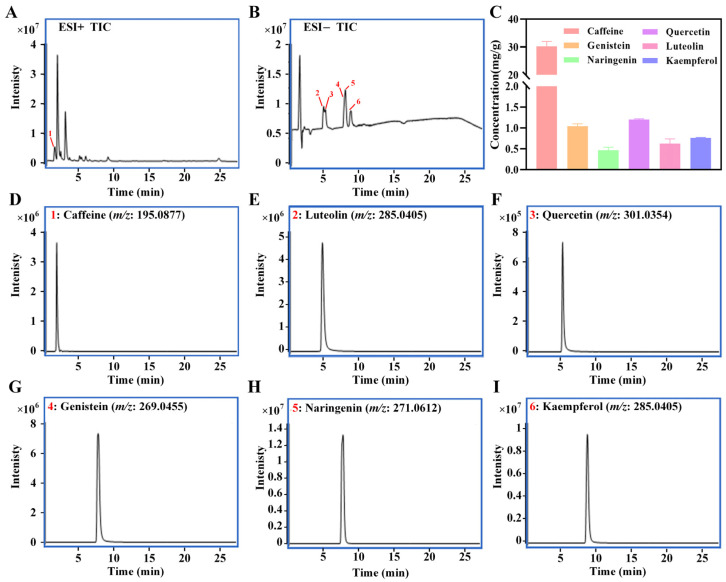
The quantification of potential ingredients in LBTE. (**A**,**B**) The total ion chromatogram (TIC) of standards detected by LC-MS in the ESI+ and ESI− modes, respectively. (**C**) The concentrations of potential ingredients in LBTE. The extracted ion chromatogram of reference standards: caffeine (**D**), luteolin (**E**), quercetin (**F**), genistein (**G**), naringenin (**H**), and kaempferol (**I**).

**Figure 3 nutrients-17-02665-f003:**
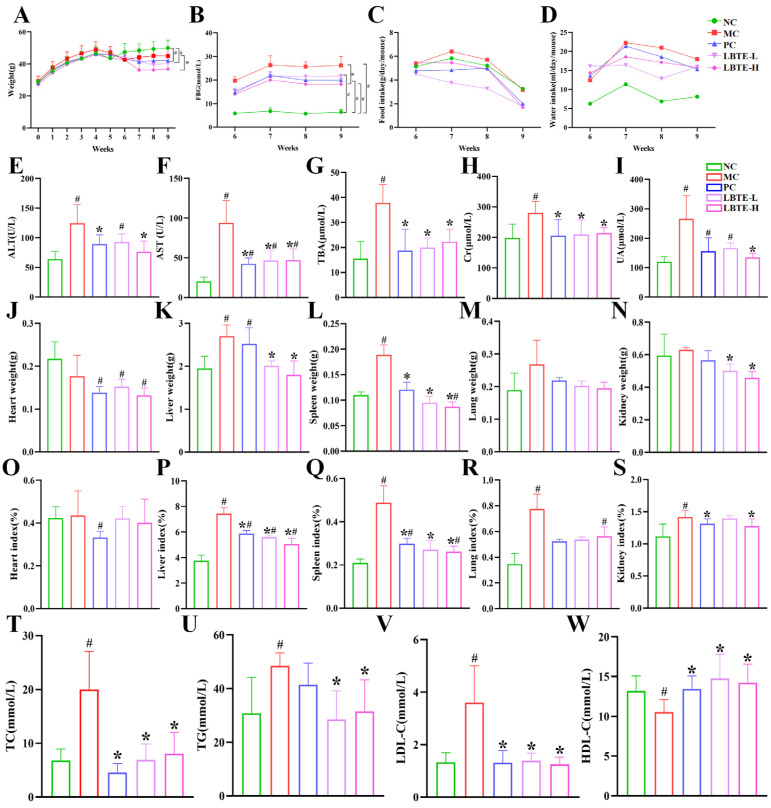
LBTE showed improvement effect on T2D mice. (**A**) Body weight. (**B**) Fasting blood glucose (FBG). (**C**) Food intake. (**D)** Water intake. (**E**) ALT, (**F**) AST, (**G**) TBA, (**H**) Cr, (**I**) UA, (**J**) Heart weight, (**K**) Liver weight, (**L**) Spleen weight, (**M**) Lung weight, (**N**) Kidney weight, (**O**) Heart index, (**P**) Liver index, (**Q**) Spleen index, (**R**) Lung index, (**S**) Kidney index, (**T**) TC, (**U**) TG, (**V**) LDL-C, and (**W**) HDL-C. NC, normal-chow diet; MC, high-fat diet (HFD); PC, HFD + Metformin (200 mg/kg); LBTE-L, HFD + LBTE (200 mg/kg); LBTE-H, HFD + LBTE (400 mg/kg). Data are expressed as mean ± standard deviation (SD) (*n* = 5). ^#^
*p* < 0.05 compared with the NC group; * *p* < 0.05 compared with the MC group.

**Figure 4 nutrients-17-02665-f004:**
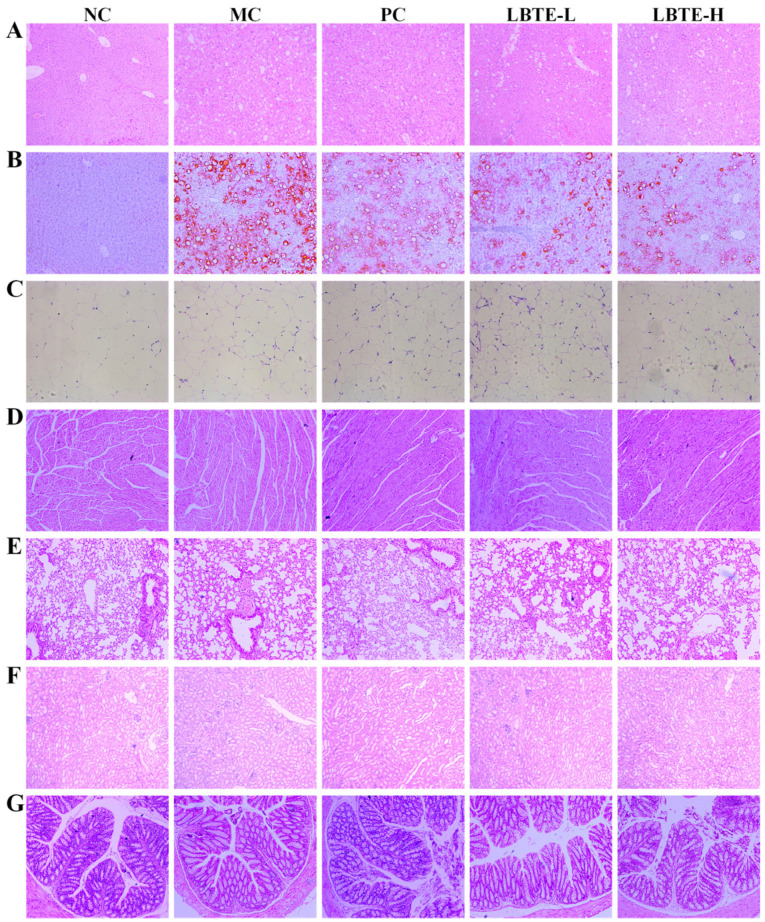
Effects of LBTE administration on the organ histopathologic changes in mice. (**A**) H&E staining of liver tissue (200×). (**B**) Oil Red O staining of liver tissue (100×). (**C**) H&E staining of adipose tissue (200×). (**D**) H&E staining of heart tissue (100×). (**E**) H&E staining of lung tissue (100×). (**F**) H&E staining of kidney tissue (200×). (**G**) H&E staining of colon tissue (100×).

**Figure 5 nutrients-17-02665-f005:**
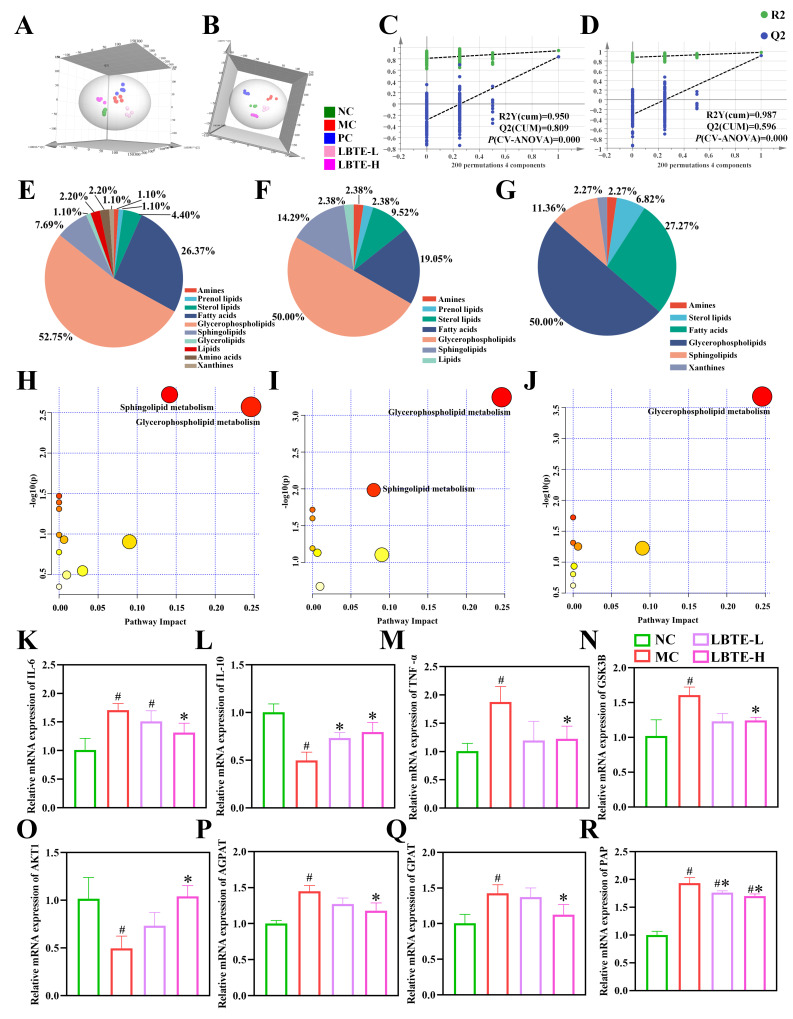
Effects of the serum metabolic profiling and the gene expression level of *IL-6*, *IL-10*, *TNF-α*, *GSK3B*, *AKT1*, *AGPAT*, *GPAT*, and *PAP* in mice treated with LBTE. (**A**,**B**) OPLS-DA of serum metabolic profiling among the five groups in both ESI modes, respectively. (**C**,**D**) OPLS-DA model permutation test for both ESI modes, respectively. (**E**–**G**) The composition ratio of serum differential metabolites in the NC vs. MC groups, LBTE-L vs. MC groups, and LBTE-H vs. MC groups, respectively. (**H**–**J**) The analysis of metabolic pathways of serum differential metabolites in the NC vs. MC groups, LBTE-L vs. MC groups, and LBTE-H vs. MC groups, respectively. The relative mRNA expression level of *IL-6* (**K**), *IL-10* (**L**), *TNF-α* (**M**), *GSK3B* (**N**), *AKT1* (**O**), *AGPAT* (**P**), *GPAT* (**Q**), and *PAP* (**R**) in the liver tissue. Data are displayed with mean ± SD (*n* = 5). ^#^ *p* < 0.05 compared with the NC group; * *p* < 0.05 compared with the MC group.

**Figure 6 nutrients-17-02665-f006:**
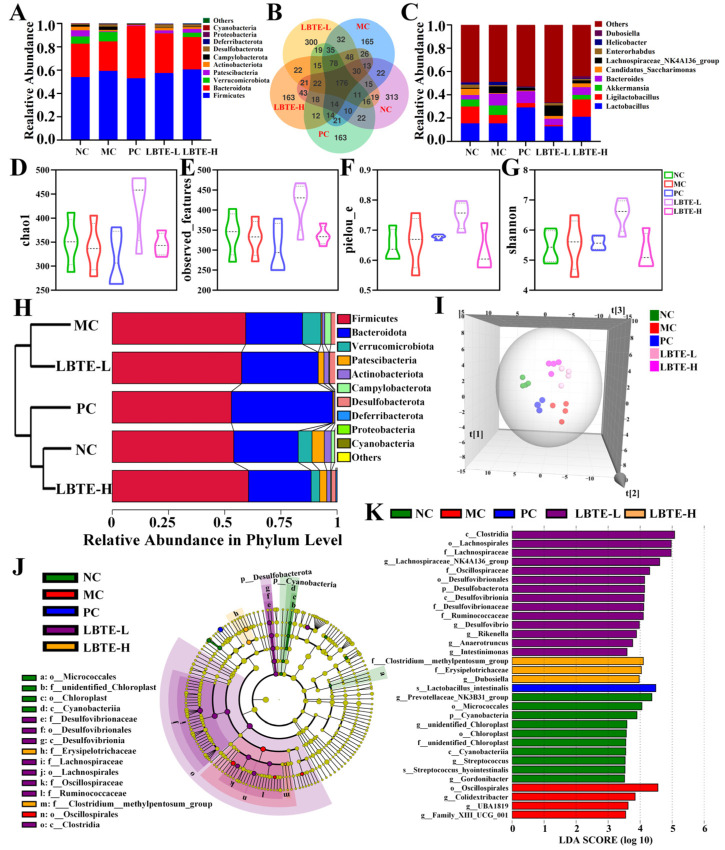
Remodeling effects of LBTE administration on the gut microbiota in T2D mice. (**A**,**C**) Relative abundance at the phylum and genus levels, respectively. (**B**) Venn diagram of ASVs identified in the gut microbiota. (**D**–**G**) Alpha diversity of gut microbiota evaluated by chao1, observed_species, pielou_e, and shannon indexes. (**H**) Hierarchical clustering tree analysis. (**I**) Betas diversity of gut microbiota evaluated by weighted OPLS-DA score plot. (**J**) Cladogram generated from LEfSe analysis. (**K**) LDA score derived from LEfSe analysis.

**Figure 7 nutrients-17-02665-f007:**
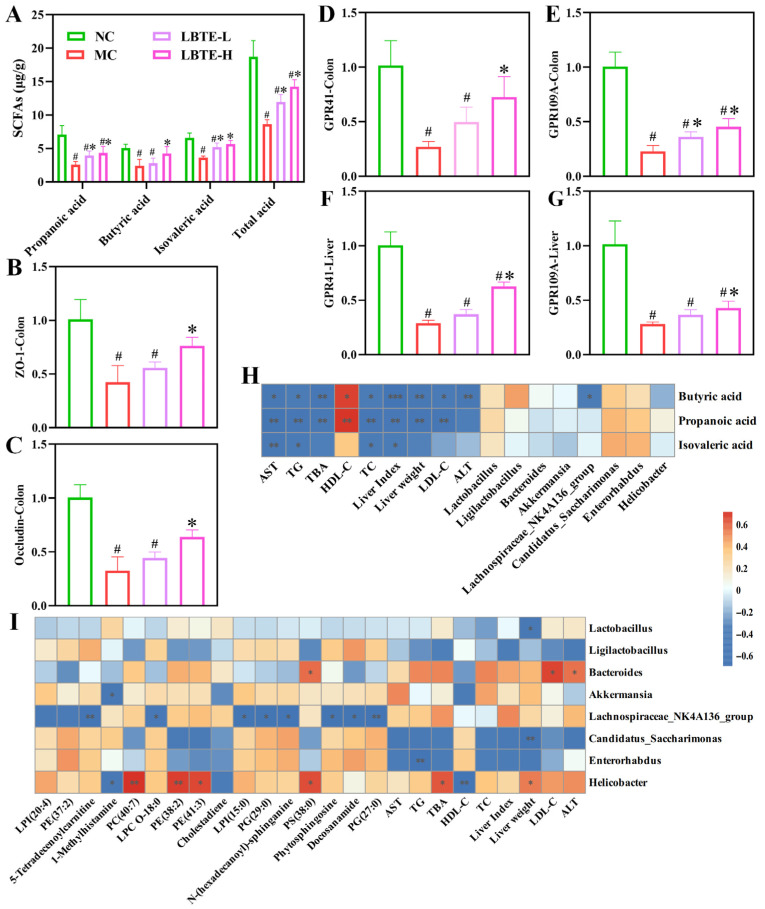
Regulatory effect of LBTE on fecal SCFAs and the intestinal barrier function in T2D mice. (**A**) The levels of SCFAs. (**B**–**E**) The mRNA levels of *ZO-1*, *Occludin*, *GPR41*, and *GPR109A* in the colon tissue. (**F**,**G**) The mRNA levels of *GPR41* and *GPR109A* in the liver tissue. (**H**) The correlation analysis between SCFAs and gut microbiota or biochemical indicators. (**I**) The correlation analysis between gut microbiota and differential metabolites or biochemical indicators. In panel (**A**–**G**), data are displayed with mean ± SD (*n* = 5). ^#^ *p* <0.05 compared with the NC group; * *p <* 0.05 compared with the MC group. In panel (**H**,**I**), significant correlations are marked by * *p <* 0.05, and ** *p <* 0.01 in the heatmap. Red and blue squares indicate positive and negative correlations, respectively.

**Figure 8 nutrients-17-02665-f008:**
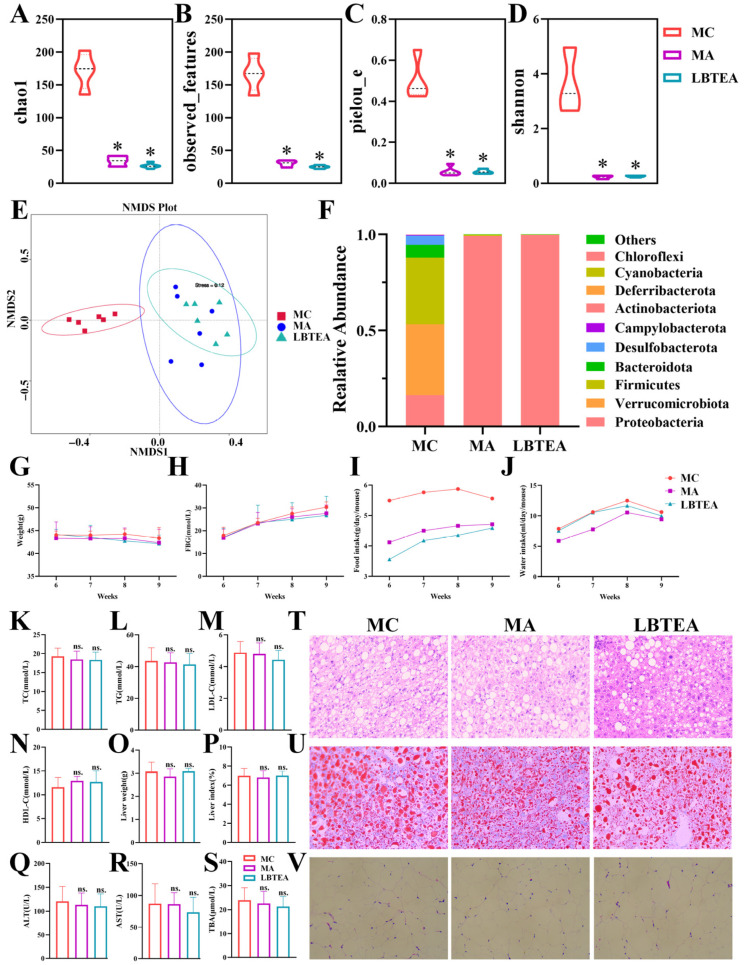
LBTE exerted improvement effect in T2D depending on gut microbiota. (**A**–**D**) Alpha diversity of gut microbiota evaluated by chao1, observed_species, pielou_e, and shannon indexes. (**E**) Beta diversity of gut microbiota evaluated by weighted NMDS plot. (**F**) Relative abundance at phylum level. (**G**) Body weight. (**H**) FBG. (**I**) Food intake. (**J**) Water intake. (**K**) TC, (**L**) TG, (**M**) LDL-C, (**N**) HDL-C, (**O**) Liver weight, (**P**) Liver index, (**Q**) ALT, (**R**) AST, and (**S**) TBA. (**T**) H&E staining of liver tissue (200×). (**U**) Oil Red O staining of liver tissue (200×). (**V**) H&E staining of adipose tissue (200×). MC, HFD; MA, HFD + antibiotic; LBTEA, HFD + antibiotic + LBTE (400 mg/kg). Data are expressed as mean ± SD (*n* = 5). * *p* < 0.05 compared with the MC group; ^ns.^
*p* > 0.05 compared with the MC group.

## Data Availability

The original contributions presented in this study are included in the article and Appendix A, further inquiries can be directed to the corresponding author.

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
