# Peer review of "Liubao Tea Extract Attenuates High-Fat Diet and Streptozotocin-Induced Type 2 Diabetes in Mice by Remodeling Hepatic Metabolism and Gut Microbiota"

_nutrients, 2025, doi:10.3390/nu17162665_

Round 1
Reviewer 1 Report
Comments and Suggestions for Authors
Luo et al describe their research into the beneficial properties of Liubao tea extract (LBTE) by determining the different components of LBTE and the effects of LBTE on mice treated with and without LBTE after HFD and STZ induction at a T2D model. LBTE-treated mice resulted in decreased body weight and reduced blood glucose values. Additionally, they had improved liver function. LBTE also altered the microbiome and removal of the microbiome by antibiotics resulted in the loss of the beneficial effects of LBTE. Overall, this is a well described manuscript looking into the beneficial effects of LBTE on T2D.
Major concern:
Did LBTE alter both inuslin production and insulin resistance in these mice?
Minor comments:
The labeling of the groups is confusing and not intuitive as the MC isn’t metformin control and PC isn’t placebo control?
Food intake appears to be decreased even if not significant, this should be a point of discussion.
Figure 5 – what tissue was the mRNA from?
Reviewer 2 Report
Comments and Suggestions for Authors
There is significant scientific value and promise in the manuscript "Liubao Tea Extract Attenuates High-Fat Diet and Streptozotocin-Induced Type 2 Diabetes in Mice by Remodeling Hepatic Metabolism and Gut Microbiota." In my opinion, as a reviewer, it requires major revision before it can be approved for publication. Its quality and clarity will be substantially enhanced by addressing the significant and minor remarks previously mentioned.
- The volume of information in the manuscript makes it challenging for readers to comprehend the main points and the story as a whole. The author needs to organise the structure, and a concise summary is necessary.
- Potential interactions between flavonoids and targets (AKT1, TNF, and GSK3B) are indicated by molecular docking. Mechanistic claims would be supported by experimental validation, such as activity assays or Western blotting for protein expression. I suggest that to author add experimental validation if possible
- The author needs to justify the choice of the LBTE dosages (200 and 400 mg/kg), including references or initial dose-finding experiments that may have been conducted.
- Need to improve the clarity of figures and tables by including comprehensive legends and ensuring consistent formatting. Breaking down complex results into summarized schematic diagrams or flowcharts could help readers better understand the proposed mechanisms.
- Need to clarify any acronyms when first introduced for reader clarity
- Proofreading is necessary to fix typographical and grammatical errors throughout the manuscript.

Round 2
Reviewer 1 Report
Comments and Suggestions for Authors
The authors have responded appropriately to the reveiwers suggestions.
Author Response
Dear Reviewers,
On behalf of all authors, I would like to extend our sincere gratitude for your valuable time and insightful comments during the review of our manuscript titled “Liubao Tea Extract Attenuates High-Fat Diet and Streptozotocin-Induced Type 2 Diabetes in Mice by Remodeling Hepatic Metabolism and Gut Microbiota.” Your constructive suggestions greatly contributed to improving the quality and clarity of our work. We truly appreciate your recognition of the study and are grateful that you found the revised version satisfactory. Thank you once again for your kind support and encouragement.
With best regards,
Qisong Zhang
Corresponding Author
On behalf of all co-authors

Reviewer 2 Report
Comments and Suggestions for Authors
Dear Authors, thank you for improving your manuscript. After reviewing carefully, I have found some points that could be a major revision if not addressed properly.
- The author needs to remove plagiarism from the manuscript, for example, page 2, lines 70-71 and 94-95.
- The current checklist nearly covers key aspects of ethical review and animal welfare, such as study design, sample size, and procedures. Nonetheless, it does not comply with one of the 3R principles: alternative animal use. However, a detailed study design and outcome measures could help determine whether alternatives were considered. To improve this, include a specific question on whether non-animal models (in vitro, computational) were considered or used.
- 2. I suggest representing the schematic diagram after the abstract, which will be more significant.
